# Facilitators, Barriers, and Educational Preparedness of Early-Career Nursing Graduates Entering Practice in Rural and Remote Areas: A Mixed-Method Study

**DOI:** 10.3390/nursrep15110410

**Published:** 2025-11-20

**Authors:** Joanne Loughery, Sai Krishna Gudi, Tom Harrigan, Elsie Duff

**Affiliations:** 1Department of Nursing, School of Health Sciences & Community Services, Red River College Polytechnic, Winnipeg, MB R3H 0J9, Canada; sgudi@rrc.ca (S.K.G.); tharrigan@rrc.ca (T.H.); 2College of Nursing, Rady Faculty of Health Sciences, University of Manitoba, Winnipeg, MB R3T 2N2, Canada; els.duff@umanitoba.ca

**Keywords:** nurse recruitment, nurse retention, rural and remote nursing, undergraduate nursing education, rural health services

## Abstract

**Background/Objective:** A nurse staffing crisis is a high-profile issue in the healthcare system. The challenge accelerates when considering the status of the nursing workforce in rural and remote (R&R) areas, where recruitment and retention are mounting problems. The primary focus of this study was to evaluate facilitators and barriers to entry into R&R nursing practice, alongside understanding educational preparedness to practice in these settings in Manitoba. **Methods:** A sequential explanatory mixed-methods survey and qualitative interviews were used as a study design to explore this emerging problem. Study participants include registered nurses (RNs) and Licensed Practical Nurses (LPNs) practicing in Manitoba’s R&R areas within three years of graduation from a nursing program. **Results:** A total of 77 nurses (56-RNs and 21-LPNs) participated in the survey, while 16 nurses were interviewed subsequently. Having a positive workplace culture (70%), being born or residing in an R&R area before practicing as a nurse (66%), and having a good clinical variety of patients (65%) were identified as key facilitators. Unmanageable workload with inadequate staffing (50%) and inadequate resources and infrastructure (46%) were identified as key barriers to entering R&R nursing practice in Manitoba. Through qualitative interpretive descriptions, the generalist role, autonomy, rural life, and organizational culture were identified as facilitators, while resources, staffing, geography, and expanded roles were identified as barriers. **Conclusions:** Preparing new nursing graduates for the realities they face in R&R areas is paramount. The current study findings help inform R&R curriculum in undergraduate nursing programs and consider strategies to enhance employment opportunities for new nurses in these dynamic settings.

## 1. Introduction

Nurse shortages are a growing concern, and the recruitment and retention of nurses are becoming increasingly challenging, particularly in rural and remote (R&R) areas. Statistics Canada defines rural areas as having a population density of less than 400 persons per square kilometre and fewer than 1000 people [1]. Contrary to the increase in rural population, the number of nurses in R&R areas of Canada is declining. Approximately 30% of the Canadian population lives in R&R settings, where only 10.8% of nurses are employed in these settings [2]. According to the Canadian Nurses Association’s recent statistics, the percentage of regulated nurses working in R&R areas has declined from 9.4% to 9.0% between 2022 to 2023 in Canada [3]. Hence, in Canada, nurse-to-population ratios are truly lower in R&R areas.

Over the last decade, the proportion of nurses working in R&R areas has significantly declined from 11.1% in 2013 to 9.6% in 2022 [4]. Several variables contribute to the nursing decline, including deteriorating workforce conditions such as increased overtime, staff shortages or mandated shift work, restructuring of healthcare systems, and increasing numbers of nurses retiring from the profession [5,6]. These variables explain why nurses leave the profession at a remarkable pace. Furthermore, the physical strain and mental and emotional stress among nurses aggravated by the COVID-19 pandemic significantly exacerbated the situation, leading to nurse shortages [7]. As a result, soaring vacancy rates (6.4% across Canada with almost close to 10% in Manitoba) and recruitment and retention of nurses in the healthcare system remain a complex and multifaceted problem [8].

According to the 2021 Manitoba Census [9], 38% of Manitoba’s population lives outside the metropolitan area of Winnipeg. There is also an increase in the number of older adults and Indigenous Peoples in these settings who frequently suffer from more complex health issues and inequities related to access to health services [10]. A recent annual population statistics regional analysis by the Manitoba Bureau of Statistics revealed that more than 43% of Manitobans live in rural and remote areas (outside the Winnipeg region), which includes more than 112,000 older adults [11]. Additionally, farm accidents, addictions, suicide rates, mental illness and chronic diseases such as heart disease and cancer are more frequently reported in these settings [10]. Considering these variables, nursing care in R&R areas remains complex. With an increasing shortage of nurses practicing in R&R settings, this remains a critical challenge in our current healthcare system.

The R&R settings are frequently required to contend with weather, geography, or travel challenges, which can be obstacles for any healthcare provider or the people in their care [12]. Rural nurses deal with challenges related to the lack of infrastructure, the need for better internet, and the lack of mental health support or limited nursing advancement opportunities, which adds an additional element of complexity to their work [13]. The unique nursing practice is a result of multiple dimensions (*‘person/al’, ‘profession/al’ and ‘place’*) and domains that influence nurses’ decisions to work in these settings [14]. Therefore, to implement a tailored recruitment and retention strategy in the Canadian workforce, key factors that inform nurses’ decisions should be considered to attract and retain nurses in these areas [14].

Considering the rapidly declining workforce in R&R settings, recruitment and retention strategies should be regarded at the undergraduate level to entice nurses to consider R&R nursing a viable and great opportunity. The primary purpose of this study was to evaluate facilitators and barriers to entry into nursing practice in R&R areas of Manitoba. A secondary purpose was to understand educational preparedness to practice in these settings. The questions and inquiries guiding this study include: *(1) What are the facilitators and barriers for undergraduate and diploma-trained nurses’ entry into nursing practice in R&R areas in Manitoba; (2) To describe the experiences of new nursing graduates’ preparedness for entry into practice in R&R areas in Manitoba; (3) To identify recommendations on promoting R&R nursing practice and enhancing educational experiences in undergraduate and diploma nursing programs.*

## 2. Materials and Methods

### 2.1. Study Setting and Design

In the current study, R&R nurses were defined as those who practiced outside Winnipeg or Brandon metropolitan areas during data collection. Also, rural was defined as communities with a core population of less than 10,000 people, where less than 50% of the population commutes to larger urban centers for work [15]. While remote was defined as communities in Manitoba that do not have permanent road access, are more than a four-hour drive from a major rural hospital or have rail or fly-in access only.

A sequential explanatory mixed-methods design was utilized, involving the inclusion and integration of quantitative and qualitative methods [16]. Combining both methodological processes provided an innovative approach to addressing R&R nursing practice preparedness and, therefore, allowed for a robust analysis utilizing the strength of each approach. The study was developed in phases utilizing sequential explanatory mixed-methods principles. In the first phase, the quantitative survey data was collected and analyzed. In the intermediate stage of the study, a data integration approach was utilized, building on the survey data results to further inform the qualitative inquiry. The study’s second phase involved qualitative interpretive description, including participant interviews and analysis. After completing the interviews, the two databases were merged for analysis and comparison (Figure 1). Study participants included registered nurses (RN) and licensed practical nurses (LPN) who worked in R&R areas of Manitoba in any practice setting and graduated in the last three years from an undergraduate nursing program. We excluded Registered Psychiatric Nurse (RPNs), NPs, and advanced practice nurses because of their scope of practice, which is different from RNs and LPNs.

### 2.2. Data Collection

The quantitative phase of the study was a cross-sectional survey. A Nursing Community APGAR questionnaire (NCAQ) was modified from 50 to 14 variables and included questions from five key categories such as (1) Geographic, (2) Economic/Resources, (3) Management/Decision Making, (4) Practice Environment/Scope of Practice and (5) Community/Practice Support was utilized in the study [17]. Each of the above-mentioned categories has ten statements (a total of 50 questions). Keeping the current study’s aim and purpose in mind, questions related to workplace culture, clinical exposure, community embeddedness, adequate compensation, access to educational support, housing availability & affordability, workload, opportunities, social & recreational activities and infrastructure were taken into consideration. A modified Nursing Community Apgar Questionnaire (M-NCAQ) with 14 statements was administered to determine the facilitators and barriers to entering R&R nursing practice. Respondents were asked to rate those 14 statements relating to R&R nursing practice on a 5-point Likert scale (strong facilitator, somewhat of a facilitator, neither a facilitator nor a barrier, somewhat of a barrier, and strong barrier). The modified survey was not piloted (pre-tested) since the reliability of the M-NCAQ was assessed by performing Cronbach’s alpha with 71 participants, resulting in an excellent internal consistency value of 0.88. Similarly, the interview guide used in the qualitative study was not pre-tested.

Sociodemographic data was collected, and two open-ended qualitative questions were included in the survey. The survey was created and later distributed using Qualtrics XM software, version March 2023 [18]. Assistance for recruitment was obtained through the College of Registered Nurses of Manitoba (CRNM) and the College of Licenced Practical Nurses of Manitoba (CLPNM), as well as advertisement across all Manitoba rural RHAs. Survey was sent out to 476 RNs via CRNM and CLPNM. Three email reminders (each with a one-month interval) were sent by the College of Registered Nurses of Manitoba. In addition, the Chief Nursing Officers sent reminder emails to their staff. Participants were entered into a draw to receive 1 of 20, $50 Canadian dollar gift certificates as an honorarium for participation.

Interpretive description (ID), a smaller-scale qualitative research design, was utilized in the study’s second phase [19]. The primary purpose of employing ID design was to identify themes and patterns within subjective perceptions to generate a description to inform practice. The detailed descriptions from the interviews were coded inductively and iteratively, which were then sorted and organized to apprehend the overall picture of the participants’ experiences. These descriptions provided a basis for informing practice and making recommendations [19]. Additionally, the participants were asked to provide feedback and insights regarding R&R nursing practice and enhancing educational experiences in undergraduate and diploma nursing programs. Then the study recommendations were generated directly from the participant qualitative interviews. Purposive sampling was utilized for the qualitative phase of the study. Data was collected from both RNs and LPNs representing all regions of the province, working in various practice settings and attending different undergraduate programs to fully capture the data broadly.

### 2.3. Participant Recruitment

Participants for qualitative interviews were recruited through survey invitation to participate, advertisement in each RHA and through snowballing and purposive qualitative sampling through the Chief Nursing Officers and word of mouth. Sixteen nurses participated in one open-ended qualitative interview from May 2023 to November 2023, where written informed consent was obtained before each interview. A qualitative analytical approach, i.e., directed content analysis was utilized in this study. Inductive analytical techniques were incorporated including repeat immersion in the data during the coding, classifying and creating linkages in the data. The principal investigator (PI) and research assistant (RA) conducted interviews via an online platform, which on average took an hour to two hours for each interview to complete. The interviews were then recorded and transcribed verbatim. The PI and RA reviewed all the transcripts, sorted and coded the data, generating ID. Data saturation and thematic adequacy was assessed when the interviewer felt new data/information no longer provides new insights or themes. Further, data collection was stopped when themes started to repeat consistently across multiple data points.

### 2.4. Data Analysis

A total of 77 participants completed the survey (out of 476 RNs; a response rate of 16.2%) and were included in the analysis. Descriptive statistics were used to summarize sociodemographic data. The perceived facilitators and barriers were ranked according to the order of significance. Since we do not have any individual cell (size) count less than or equal to 5, a Chi-Square Goodness of Fit (χ^2^) was performed on each of the 14 test items to determine if the proportion of perceived facilitation and/or barrier was equal. A Benjamini–Hochberg Procedure (B-H) was employed to control familywise error using a false discovery rate (FDR) of 0.05. Reliability of the M-NCAQ was assessed by performing Cronbach’s alpha with 71 participants, resulting in an excellent internal consistency value of 0.88. Keeping practical reasons in mind, and since reliability was already over the desired value, i.e., 0.88, a realistic sample size of 77 participants was recruited into the study. Data was entered and analyzed using SPSS version 29.0.2.0. Upon performing content analysis, the results of the quantitative survey informed to develop the nature and type of questions for the qualitative phase of the study.

## 3. Results

### 3.1. Quantitative Phase

The final sample for the quantitative phase consisted of 77 participants. Most participants were married (57.1%), belonged to the Southern Health region (29.9%), had an educational RN degree (72.7%), employed in hospital settings (64.9%), lived in a rural setting in Manitoba (68.8%), for more than 15 years (48%), respectively (Table 1).

### 3.2. Facilitators and Barriers

Identifying facilitators for R&R nursing practice is important to address the Canadian health workforce crisis. More than three-fourths of the study participants (76.1%) found that having a positive workplace culture (e.g., recognition, flexible shifts, collegial relationships) is a facilitator (Figure 2). While 71.8% of the participants agreed that having been born or resided in an R&R area before practicing as a nurse was a facilitator. Having a good clinical variety of patients is a strength of remote nursing. In addition, almost two-thirds of respondents (60.6%) found Community Embeddedness (e.g., perception of quality, sense of reciprocity with community, image of rural healthcare) to be a facilitator. More than half of the study participants (56.3%) found being educated in an R&R area through practicum or training at a satellite site to be a facilitator. Similarly, more than half of the study participants (54.9%) found that having a robust orientation or welcome program and adequate total compensation (e.g., relocation allowances, differentials, overtime, benefits, salary, flex scheduling) to be facilitators. Similarly, more than half of the study participants (53.5%) found being able to be involved in healthcare decisions and access to continuing education support (professional development opportunities, scholarships, remote learning) to be facilitators. Lastly, almost half of the study participants (49.3%) found the affordability and availability of housing in R&R areas to be a facilitator, although it was not statistically significant (Table 2).

More than half of the study participants (53.5%) found having unmanageable workloads and inadequate staffing to be a barrier. Almost half of the study participants (49.3%) found having inadequate resources, infrastructure, and informatics (e.g., electronic charting, usable equipment, internet, emergency medical services) to be a barrier. However, it was not statistically significant (Table 2). In addition, having limited access to opportunities for their families (e.g., childcare/education, spousal employment) (45.1%) and having limited social and recreational opportunities (45.1%) are barriers, respectively (Figure 3). R&R nurse barriers to practice are useful to identify and address to recruit or retain these experts.

### 3.3. Qualitative Phase

Sixteen nurses employed across all four rural RHAs in Manitoba were interviewed for this study, including RNs and LPNs (Table 3). Four nurses were employed with an agency service and worked in multiple settings and regions of the province, including Winnipeg RHA. Participants who had experience working in urban and R&R settings provided a unique comparison of urban/rural nursing practice. Three participants lived in an urban/metropolitan area and commuted to their workplace outside the city limits in a rural or remote hospital. While the remaining participants resided in their rural or remote settings, and some had to commute to their place of employment. Most nurses were employed as general duty staff nurses in a hospital setting; however, differences in experiences exist. We interviewed nurses working in small community hospitals and larger hospitals in bigger cities across the province. A variety of experiences were explored, such as general medicine, surgery, geriatric, long-term care, palliative care, pediatrics, obstetrics, and emergency nursing. Nurses were engaged in various roles, from front-line service to overseeing entire hospitals, where one nurse was in a manager position. Most of the study participants were from the undergraduate and diploma nursing programs, while five nurses obtained their training outside the province of Manitoba (Table 1).

### 3.4. Qualitative Interpretive Descriptions and Themes

Findings from this study phase have been categorized as barriers and facilitators to R&R nursing practice. Common patterns emerged in the data provided descriptions of the first few years in the nursing practice. The participants were asked to provide insight into a series of recommendations for practice and education, which is featured. Individual experiences in the context of a particular circumstance or unique practice settings were captured in the data, and commonalities shared by many participants were also featured [19].

### 3.5. Barriers

#### 3.5.1. Resources

Most participants voiced concern over the lack of resources such as equipment, technology, and diagnostics compared to the urban setting. There was concern about outdated facilities, non-functioning equipment, reliance on faxing or other technologies, and long delays in obtaining results that waste time in their daily work life. Consultations with physicians or other disciplines also varied, creating service gaps. Participants with urban experiences described the difference between resources and interdisciplinary support in a rural setting as significant. Having specialized teams and readily available resources or equipment created a different experience in urban settings. Managing these challenges in rural settings was described as a significant learning curve and many felt unprepared as a new graduate. Challenges related to resources remained consistent across practice settings in various regions of the province. However, the concerns were accelerated in smaller hospitals or isolated areas.


*“I realized I am really on my own up here and I need to get creative with how I do things … No doctor on site, limited labs, and diagnostics…”*

*RN Northern Regional Health Authority*


Nurses also recounted experiences where a lack of resources created concerns for patient outcomes. When a seriously ill patient arrives in a facility without adequate capacity, the nurses are the first and, in some cases, the only point of human contact on-site, leaving the nursing staff in a precarious situation. Nurses raised concerns about feeling unprepared to handle these urgent experiences directly from graduation.


*“It does not usually end in a great outcome for the patient. If we don’t have the resources to help them, they can’t get out. It can be very scary.”*

*RN Prairie Mountain Regional Health Authority*


#### 3.5.2. Staffing

Staffing was a barrier identified by most participants across all settings. Understaffing, mandated overtime, long hours of work, and high vacancy rates were reported as frequent problems in many regions of the province. Additionally, staffing challenges also put many in a situation where they did not have access to expertise or senior staff to help them navigate unfamiliar patient experiences. While urban settings face similar staffing issues, these challenges can be amplified in R&R communities based on a range of problems such as location, weather, recruitment and retention in the regions.


*“Trying to get staff to come to the rural area is a big challenge not only for nurses but for the whole care team. Having all those staffing issues comes with having to work doubles and being mandated.”*

*RN Prairie Mountain Regional Health Authority*


One participant talked about closing certain departments in the hospitals due to a lack of staff, while other participants reported emergency departments reducing their hours or closing completely based on staff availability. Many participants voiced concern over patient outcomes, higher acuity situations, or transporting patients to an alternative setting due to staffing challenges.


*“Our observation unit is closed in the E.R. because we don’t have the staff to keep the four beds open. So, I feel like it’s kind of domino effect and just overall not very good for the patients seeking care.”*

*RN Southern Regional Health Authority*


Another recurring theme from the interviews was the reliance on agency nurses, especially during staff shortages. One agency nurse who floated to a remote fly-in setting explained that the nursing station is entirely staffed by agency nurses which left them in unfamiliar situations. There were mixed views on working with agency nurses, where, mostly, nurses were happy to have extra help. Others voiced concerns over working with agency nursing staff, such as lack of consistent care, familiarity with the unit, patient needs or hospital policies, or they would need to assume additional responsibilities, such as being in charge when hospitals are staffed by agencies.


*“I think it’s a great thing that these agency nurses come and help us out … But it’s like you’re orienting a new person every single day. They don’t understand the same policies …or they don’t know where stuff is … So sometimes you’re better doing things yourself and that can get frustrating.”*

*RN Prairie Mountain Regional Health Authority*


In addition, the agency nurses who were interviewed expressed mixed views, where, often, they enjoyed working in a rural setting as they felt it was a break from the chaotic, fast pace of an urban hospital. That is, these agency nurses enjoyed being able to spend time with and provide more holistic care to the people in their care. However, agency nurses (*n* = 5) expressed that they did not feel being a part of the team and often felt undervalued by the other staff.

#### 3.5.3. Geography

The geographic need for ground or air patient transportation created challenges for the R&R nurses. Patients often must be transported to urban centers for medical appointments, procedures, specialized care, diagnostic imaging, or emergency care. Ambulance services often were delayed due to weather, traffic or road conditions, which directly affects patient care and related outcomes. Fly-in settings also posed similar challenges, where one nurse in a remote setting shared that waiting two to three days for a flight patient transfer was a frequent occurrence. These geographic challenges would often put the nurse in a position where they may have to care for unstable patients, or they were not equipped to handle patients with increased acuity. Nurses also described a learning curve with the complexities around patient transport and felt unprepared to arrange transportation out of graduation from their nursing program.


*“Geography is always a challenge; we still get quite severe storms and that shuts the highways down … and you have to stabilize a patient for 48 h before the roads open back up so you can get them out.”*

*RN Prairie Mountain Regional Health Authority*


Staff having to commute to work, supplies needed to be brought in, or patients travelling for appointments or diagnostic tests would also have a direct effect on the flow and safety of the hospital. The nurses said they frequently had to juggle patient services, bed utilization or patient discharges based on geography and distance-related challenges.


*“When there’s weather and things that you can’t predict, patients don’t get to go to their appointments … which is the last step for them going home, it’s very frustrating.”*

*RN Northern Regional Health Authority*


#### 3.5.4. Expanded Role

The nurses also raised that they often take on additional roles beyond their nursing responsibilities. These roles may include tasks typically performed by physiotherapists, respiratory therapists, social workers, occupational therapists, or other healthcare professionals. One nurse described that after hours, nurses were the only staff in the hospital, and they would take on the role of healthcare aid, housekeeper, or laundry services. They often needed to perform tasks such as drawing lab work or blood gases, assisting with electrocardiograms, and handling other equipment they were never trained to use. Nurses described that they were often required to be adaptable and provide care beyond traditional nursing roles.


*“If we’re going to see a patient in the ER after 4, for and emergency, the nurse would do the EKG, and we would draw lab work as well.”*

*RN Southern Regional Health Authority*


The nurses were often assigned leadership roles, such as overseeing units or managing staffing issues, shortly after obtaining their nursing license. Some described working entire shifts with agency nurses and were the only experienced staff in the hospital at times. One participant described being placed in charge of the hospital within months of graduation, while others described it as standard of practice within the first year. Many felt unprepared and uncomfortable when placed in charge early in their careers as they lacked the necessary training and experience to handle these responsibilities.


*“It felt horrible being in charge … I was there for 8 months and there were no RNs on shift, and I was the most senior LPN so I was charge by default and it’s stressful to manage everything … I was not prepared.”*

*LPN Interlake Eastern Regional Health Authority*


Additionally, when other disciplines were not available, such as physicians on site, nurses were often required to make independent decisions while consulting with the physician from afar. This steep learning curve directly after graduation was stressed by many of the participants interviewed. This expanded role and increased responsibility was a significant R&R practice barrier they frequently juggled.

### 3.6. Facilitators

#### 3.6.1. Generalist Role

Nursing in a rural or remote area was often described as a generalist practice and viewed positively. Day-to-day practice involved being able to care for a variety of patient situations which required a broad knowledge base. R&R nurses were often required to be flexible and adaptable to any situation. However, this was not always the case in some of the larger rural hospitals in the province where specialized medical or surgical units exist, which would be more consistent with an urban setting. However, in smaller hospitals, generalist nursing practice is a typical occurrence, where most of the nurses interviewed viewed this as a positive aspect of their job and enjoyed the learning and range of experiences.


*“I do like the opportunity to see a lot of different things … the medicine unit is combined palliative care … I think that’s cool that I can kind of practice two specialties and being able to see so many different … Systems, like cardiac GI, GU, on the same medicine unit.”*

*RN Southern Regional Health Authority*


Most of the nurses interviewed considered the variety of opportunities to be a significant facilitator to their nursing practice. Many talked positively about daily learning because of generalist practice and felt they were becoming more prepared, based on these experiences, to be able to work in any practice setting. The agency nurses who worked in urban and rural centers described patient diversity with a positive lens. The nurses mostly enjoyed spending time in the rural setting, as it was viewed as a pleasant change to the chaotic and specialized care they were used to in an urban setting.


*“I like the diversity of patients … it’s ever changing. There’s always something new going on. I feel like I’m always learning every day. And I think it’s been fantastic … I wouldn’t necessarily be doing that if I was in Winnipeg unless I was in a float position.”*

*RN Northern Regional Health Authority*


#### 3.6.2. Autonomy

The nurses described their role as requiring a heightened level of autonomy. Clinical decisions were invariably affected by many factors, including access to resources, health services, on-site medical staff, available diagnostics and other service disciplines. The nurses were often required to be flexible and creative and required to make critical decisions while collaborating with the physician from afar. The increased collaboration, independence and autonomy fostered a positive working relationship that most nurse participants enjoyed.


*“I do love the autonomy of rural nursing. I feel like I’m very close with the doctors. They trust our nursing judgment. We create plans together, and they respect our opinions.”*

*RN Prairie Mountain Regional Health Authority*


Collaborating and close working relationships with physicians were a vital aspect of the role, as described by most nurses who were interviewed. The nurses felt that the trust physicians have in them was very important. They felt supported and comfortable making independent decisions or utilizing appropriate standing orders as necessary to make critical patient care decisions. It was also expressed that the physician’s mentorship and teaching role was important early in their career, and the trust and respect that developed over time was positive. Fostering these relationships supported an increase in the autonomy and confidence of the nurses in practice.


*“The physicians are eager to kind of teach and you can ask any questions. I feel like the relationship is pretty good with the physicians.”*

*LPN Southern Regional Health Authority*


#### 3.6.3. Rural Life

Many participants choose to practice nursing in these communities because they love rural or remote living. Community ties, family, relationships with colleagues, decreased cost of living, peaceful/quiet atmosphere, work–life balance and outside recreational activities were some of the positive descriptions that were captured. However, some nurses were planning to move into an urban setting to expand their career opportunities at some point in the future. The nurses planning on staying in a rural setting were often born and lived in the area for most of their lives.


*“Rural nursing is for me. I like it. Because it’s a small community, and I like it because it’s peaceful and quiet. It is less stressful than living in the city. Our cost of living is not that so expensive so I can work less.”*

*RN Southern Regional Health Authority*


We also explored potential concerns around caring for neighbours and the lack of anonymity in the setting based on the nurses’ position in the hospital. Most nurses were unconcerned with this and could maintain professional boundaries due to their undergraduate training.

#### 3.6.4. Organizational Culture

A flexible and supportive manager fostered a positive workplace environment where the participants appreciated managers who stood up for and supported them. They were also flexible in scheduling and attentive to their needs outside of work. Many of the nurses talked about teamwork and positive working relationships with staff including physicians, and many became friends with their colleagues inside and outside work.


*“I would say that the overall kind of teamwork and camaraderie much better in the North.”*

*RN Northern Regional Health Authority*


Nurses who worked in both urban and rural settings commented that working in a remote or rural rotation was like a vacation because they felt appreciated, had decreased stress levels, and had better work–life experience overall. They frequently reported that the patient/nurse ratio and level of acuity were often improved, providing them with the ability to provide more holistic care and spend more time with patients and their families.


*“I worked 2 days in Winnipeg, and I am exhausted and drained. They appreciate us so much more Northern and I’m on day 5 of 7 and feeling like I still have so much energy.”*

*RN Northern Regional Health Authority*


## 4. Discussion

This study aimed to identify barriers, facilitators, and educational preparedness of graduate nurses entering practice in R&R areas of Manitoba. A nurse staffing crisis accelerates when considering the status of the nursing workforce in R&R areas, which became a main finding of this study. In Canada, the proportion of regulated nurses in R&R areas continues to decline compared to the regional share of the population, where the regulated nurses leaving the profession is slightly higher in R&R areas (7.4%) than in urban areas (5.9%) [20]. Canada’s aging rural nursing population is consistent and almost similar to other countries; however, the lower proportion of rural nurses in younger age groups is now a major concern. Consequently, it is essential to have a rural workforce of nurses who can provide optimal healthcare services. The study findings contribute to a body of literature on rural healthcare issues and provide a foundation for nurse educators and healthcare administrators to address key practice areas.

### 4.1. Overview of Rural Nursing Practice

Rural nursing is often described as a generalist practice. Unique generalist descriptions of rural nursing practice are frequently reported in the literature and are consistent with the findings of this study. R&R nurses must have a broad range of knowledge and critical thinking skills [21,22,23]. Moreover, R&R nursing requires skills in multiple specialist roles (i.e., care provider, leader, educator, and advocate) to address patient care needs. Rural nurses enjoy clinical variety while utilizing multiple skills and drawing on many experiences [13,21,24]. Throughout a shift, R&R nurses may see patients who need acute, chronic, rehabilitative and/or primary care services [22,23,25]. Additionally, working in R&R areas can be difficult, where geographical isolation poses challenges through working beyond their professional capacity, adapting to extreme weather conditions, limited access to education or professional support, and patient safety concerns.

Due to the shortage and decline in rural healthcare providers, RNs working in R&R communities require additional staffing support to address complex inpatient care [26,27]. It became evident that reliance on agency nursing is a frequently occurring staffing alternative in both R&R areas of Manitoba. Additionally, nurses in the study were often taken on multiple roles in other health disciplines in many settings. The blurring of professional roles is frequently documented in the literature consistent with R&R nursing practice [13,21,24]. R&R nurses are required to adjust to a heightened level of autonomy, which is a frequent occurrence in practice [21,22]. Along with increased autonomy, R&R nurses are often required to perform to the full extent right after graduation [22,23,25]. Many of these unique attributes of rural nursing practice were identified as barriers and facilitators and should be considered when preparing graduates in undergraduate nursing programs or new graduates entering practice in these settings.

### 4.2. Barriers to Rural Nursing Practice

R&R nursing is a complex yet rewarding practice area; however, it must be considered carefully as there are challenges associated with practicing in such settings. Staffing challenges and increased workload were the barriers identified in the study that were most frequently reported. Recruitment and retention challenges, staff shortages, and high turnover rates, especially in R&R areas, continue to be significant challenges across Manitoba [28]. In addition, a national survey conducted in R&R areas in Canada noticed that the proportion of the rural nursing workforce in Canada continues to decline in relation to the proportion of the Canadian population in R&R areas. In addition, the casualization of the nursing workforce has been a continuing concern in Canada, particularly in rural areas [26]. Additional barriers were also noted in the study findings, which were related to inadequate resources, limited diagnostic services, lack of infrastructure, closures of emergency services, transport challenges, and isolated work environments.

The provision of nursing care in R&R areas is considerably more complex than in urban areas due to isolated work environments and limited access to resources, which is also noted in the literature [26,29]. Considering the aging and growing population in our country and the demand for additional health services, it is imperative that the critical nursing shortages and management of resources in these settings are addressed. According to the Government of Canada [28], several key initiatives and investments have been addressed to account for the demand for services. These include the implementation of an Undergraduate Nurse Employment Program, the creation of additional nursing education seats in all nursing programs across the province and the allocation of funds for additional health programs and facility expansions. While these are important steps in addressing this important healthcare concern, under-resourcing and understaffing in R&R settings remain a common and challenging barrier.

Interestingly, an integrative review explored common challenges and stressors experienced by R&R nurses that were not a significant concern to those nurses who had an existing emotional connection with the community [30]. Community embeddedness, residing in rural areas and love of rural life were important facilitators featured in the study findings. Therefore, an opportunity exists in undergraduate nursing programs to promote rural nursing by providing insight into the common barriers associated with rural nursing practice so that new graduates are better equipped upon graduation.

### 4.3. Facilitators to Rural Nursing Practice

Despite the many challenges that rural nurses face on a day-to-day basis, many facilitators were also featured in the study findings. It became apparent that fostering a positive workplace culture, providing a sense of belonging and a collegial atmosphere, allowing nurses to work to their full scope and developing expert generalist nursing skills should be driving factors for recruitment in R&R areas. Featured in an umbrella review, additional interventions for supporting nurse retention in R&R areas included enhancing communication and information technologies support, promoting career pathways in rural health, fostering supportive relationships in the workplace, and enhancing financial incentive programs, which also proved to improve nurse retention in R&R areas significantly [31]. This overview of systematic reviews also focused on personal and professional support, education and continuous professional development interventions influencing nurse retention in R&R areas. The findings of this study emphasized the value of continuing education, scholarship remote learning opportunities and robust orientations as key facilitators to promote practice.

Additionally, the study by MacLeod and colleagues revealed additional incentives such as interest in practice settings, flexibility of work, location of community, spouse employment/transfer, income, lifestyle, benefits, advanced practice opportunities and career advancement and continuing to work in the primary work community by region of primary employment. However, the ranking of these factors varied across regions and nurse types. Improving recruitment and retention of R&R area nurses must be a priority other than just offering financial incentives or remuneration and associated entitlements. In addition, maintaining social and cultural connectedness and working within functional teams becomes particularly valuable to rural, remote area nurses [26]. Indeed, early-career registered nurses are the future of the nursing workforce, with entry into practice at the undergraduate level at the forefront; therefore, their experiences, needs and preferences should be given priority to improve and maintain a sustainable rural nursing workforce.

A qualitative descriptive study that explored the experiences of early career registered nurses in rural hospitals in Australia further identified strategies they believe would help increase job satisfaction and retention, noticed that strategies such as social gatherings to enhance connection, sufficient orientation, greater involvement in the choice of rotations and clinical areas, desire for more flexible work hours, and assistance with accommodation and transport would help rural nurses to overcome challenges in their roles [32]. Adding incentives to facilitate nursing practice in these settings is a key strategy that nursing administrators and policymakers should consider.

### 4.4. Transition into Nursing Practice

Transitioning into nursing practice after graduation is a complex process. Adding the nuances of rural or remote nursing practice adds an additional element of complexity to the process. Duchscher describes the process of entering practice after graduation as a ‘*transition shock*’ [33]. New graduates cope with many emotions such as confusion, disorientation, doubt, and loss … Balancing workload, attaining required skills, navigating various healthcare roles, establishing collegial relationships, assuming increased responsibility and managing organizational pressures are new competencies for all nursing graduates and early career nurses. The nurses who participated in this study voiced many challenges with the transition process based on the added complexity of rural nursing practice. Educators and administrators must provide new graduates with the information and support needed to navigate the process in the rural context.

New nursing graduates’ transition to employment is a notable change and often challenging, which could lead to a phase of transitional shock. Interestingly, a longitudinal convergent mixed-method parallel design study that determined whether the current rural graduate programs in Western Australia adequately support new graduate nurses transitioning into practice concluded that new graduates experienced both transition shock and the honeymoon periods on commencement of employment, reporting high levels of satisfaction along with signs of transition shock [22].

New nursing graduates must navigate the system with increasing responsibilities, roles, knowledge, and relationships. When selecting employment opportunities, new graduates will need consistency, predictability, familiarity, and stability to be successful in the transition process and into the first few years of nursing practice. It became clear in the study findings that fostering a positive, supportive and collegial atmosphere in the work setting was a clear facilitator to practice therefore administrators must consider transition interventions as a recruitment and retention strategy.

### 4.5. Educational Preparedness of Newly Graduated Nurse

As nursing educators, additional educational needs of nursing graduates who wish to choose R&R nursing as a career opportunity should be considered. It became clear in the study findings that participants felt ill-prepared for a variety of experiences unique to R&R nursing. Therefore, the inclusion of rural nursing practice methodologies and experiences in the undergraduate nursing curriculum is imperative. Patterson and colleagues completed a scoping review to explore effective clinical education models suitable for rural nursing education, which could be considered in undergraduate nursing programs [34]. Eighty-two studies were included in the review, and clinical models were categorized based on three features, including mentorship/clinical supervision, type of care provided and placement design. The authors concluded that opportunities exist for undergraduate nursing programs to develop partnerships with a variety of rural health organizations to provide learners with the opportunity to learn in, with and about rural health and healthcare. It would expose students to the realities of rural nursing practice in these often-under-resourced settings [34].

Attracting graduates to rural nursing practice as a viable career choice should also start at the undergraduate level. A cross-sectional Australian study that explored nursing students’ decision-making concerning future rural healthcare employment proposed a Rural Nursing Workforce Hierarchy of Needs model [35]. This model offers an understanding of what nursing students consider to be fundamental needs in planning their rural employment. This hierarchical workforce needs model has six different components: clinical, managerial, practical, fiscal, familial, and geographical. In short, this study’s findings suggest that clinical needs must be met for nursing students to consider a career in a rural healthcare setting, followed by other components. Including clinical education models, rural nursing practice theory and other such awareness strategies is crucial to improving the long-term stability and sustainability of the rural nursing workforce.

A longitudinal qualitative follow-up study conducted in Western Australia that looked at personal and professional decision-making around rural nursing practice intentions and subsequent rural employment and retention suggested that among various factors that facilitate nurses’ rural practice intention, employment and retention, having a positive experience in rural placement gives students, as well as graduate nurses, the confidence to be able to pursue their careers close to rural areas [36]. On a particular note, this paper highlights the aspects related to confidence among nurse graduates who are working in rural areas. Interestingly, the study revealed the impact of rural undergraduate placements on personal and professional growth, particularly in boosting confidence in accepting rural roles and broadening available career options.

Alongside enhancing confidence, moving away from home and accepting rural roles would potentially improve knowledge, clinical skills and networking. In addition, a recent scoping review that attempted to summarize the available evidence on clinical nursing education models in rural areas found a diverse variability in rural clinical learning opportunities for nursing students [34]. It concluded that clinical education models, such as academic practice partnerships between educational institutions and rural communities, will help nursing students excel in rural clinical education.

A descriptive phenomenological study that explored early career nurses’ experiences during their first year of rural practice revealed that the effect of their vulnerability entering a new workplace and the importance of connection to person, place, and profession had influenced their decision-making to remain in rural employment [37]. The early-career nurses in the study also highlighted factors that influence their career location decisions, i.e., whether they would stay in or leave their rural employer. Establishing connections with a person, a place, and their profession can help nurses when entering a new workplace. The findings from a constructivist grounded theory that explained the turnover intention among early-career nursing professionals working in rural and remote Australia inferred that an early-career nurse’s decision to stay or leave their job was determined by the meeting of life aspirations [38]. Furthermore, it depends upon the extent of the gap between individual nurses’ professional experiences (such as job role, workplace relationships and level of access to continuing professional development), personal satisfaction, the reality of their current employment and living experience in rural areas. A recent scoping review that explored the factors that nurses practicing in rural, remote and isolated locations consider important for attraction and retention highlighted approaches that were required to plan, which could facilitate ways to address nurses’ specific needs and experiences for the future nursing workforce [39]. Concerning attraction, retention and resignation, this study further identified themes such as demanding role and scope of practice; values divergence and professional opportunities; continuing professional development and mentoring; social, lifestyle and personal or family; management and organization; and pay and incentives.

A recent qualitative study that explored the experiences of new graduate registered nurses in caring for deteriorating patients in rural areas concluded that they were unprepared to care for these patients [40]. This study further stated that practice support and barriers to ongoing education are the factors affecting their experiences. These findings highlight the need for focused rural healthcare preparation along with structured practice support through senior rural nurses and health facility orientation programs. A similar qualitative study that attempted to identify influences and drawbacks for early career nurses to work in a rural location stated that “rural early career nurse employment; proximity to social and/or familial ties, being attracted to rural clinical practice, taking advantage of a job offer in a limited market, and wanting a rural lifestyle” as influencers [41]. While “distance from social and/or familial ties, rural lifestyle factors, resource challenges, and a perception of less professional opportunity” as drawbacks. The results of this study concluded that ‘proximity to social and/or familial ties’ as both an influence and a drawback for early-career nurses to accept employment in a rural setting. Another similar study that looked at the practice readiness of the early career registered nurse in their first five years of practice within a remote hospital felt not ready for employment in a remote setting [42]. Further, later career registered nurses opined that critical care placements in an emergency or intensive care unit are needed for preparedness in being employed in R&R settings. They also stated that organizational support through appropriate orientation, supernumerary time, and adequate education was lacking.

### 4.6. Strengths, Limitations, and Future Directions

A major strength of the current study is the use of a sequential explanatory mixed-method research design that generated broader insights and a deeper understanding of a multifaceted phenomenon while utilizing the strengths of both qualitative and quantitative methods. Additionally, this is a worthy, timely, and relevant topic, considering the challenges in our current healthcare systems across the country. Preparing nursing graduates to enter practice in R&R settings requires critical consideration. Furthermore, recruitment and retention strategies in R&R practice settings are paramount. The recommendations listed below have been generated directly from interviews with the participants and provide additional considerations for this emerging problem. Other innovative ideas like satellite campuses in rural communities, increasing total compensation, and increasing facility support are all avenues to explore further. Since the study participants are early career nursing graduates, representative of a nursing profession, there might be potential for response or social desirability bias. However, since we employed anonymized de-identified data collection, maintaining privacy and confidentiality of the recorded responses, and the use of open-ended questions during the qualitative interviews, the occurrence of such bias is minimized. The data collected and analyzed in this study is Manitoba data, which could be considered a limitation to the study findings and recommendations. These findings could be difficult to generalize to other practice settings or educational facilities due to variations in health services nationwide. Therefore, future research at a national level (a Canadian study) is warranted. Having a lower sample size of 77, findings should be read with caution while drawing inferences. Lastly, besides having R&R placements in training, specific educational strategies in collaboration with other academic institutes and healthcare services should be further explored.

### 4.7. Recommendations

Undergraduate Nursing Curriculum:Increase capacity for rural placements in all clinical courses;Increase the threading of R&R content throughout theory courses;Consider the creation of a dedicated rural nursing elective to provide more exposure to R&R complexities in nursing;Facilitate R&R tours or rural clinical placements for nursing students who may wish to consider this career choice;Expand satellite nursing campuses so nurses can learn where they live and will eventually practice;Expand opportunities for final practicum placements in R&R settings;Increase partnerships with First Nations Communities and consider clinical opportunities in these settings.Practice:Extend monetary and housing incentives for remote nurses;Improve retention bonuses for nurses seeking employment in these settings;Expand the provincial nursing float pool system;Develop more robust orientation and preceptorship programs for new graduates;Consider self-scheduling and increase flexibility in work life for staff;Create a positive workplace culture that encourages new graduates to work in these settings;Expanding continuing education opportunities and subsidizing the cost of additional training in an urban setting.

## 5. Conclusions

Preparing new nursing graduates for the realities they face in R&R areas is paramount. This study revealed the facilitators, barriers, and educational preparedness of undergraduate- and diploma-trained nurses entering nursing practice in R&R areas in Manitoba. Having a positive workplace culture, being born or residing in an R&R area, and having a good clinical variety of patients were identified as key facilitators, while unmanageable workloads with inadequate staffing and inadequate resources and infrastructure were identified as key barriers to entering R&R nursing practice in Manitoba. It also proposed real-time recommendations for promoting R&R nursing practice and enhancing educational experiences in undergraduate and diploma nursing programs in Manitoba. The current study findings help inform the nursing curriculum in undergraduate nursing programs in Canada, improve nursing education, promote R&R nursing as a practical and exciting career choice, and enhance employment opportunities for new nurses in these dynamic settings.

## Figures and Tables

**Figure 1 nursrep-15-00410-f001:**
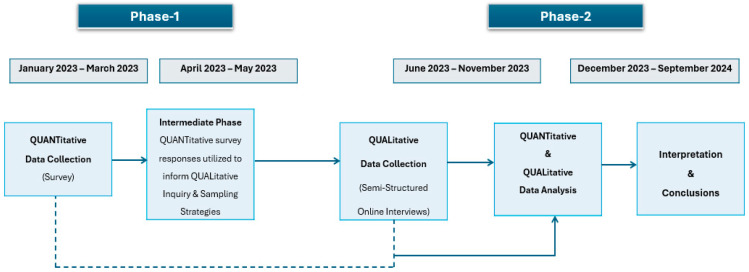
Timeline and process flow of data collection and analysis.

**Figure 2 nursrep-15-00410-f002:**
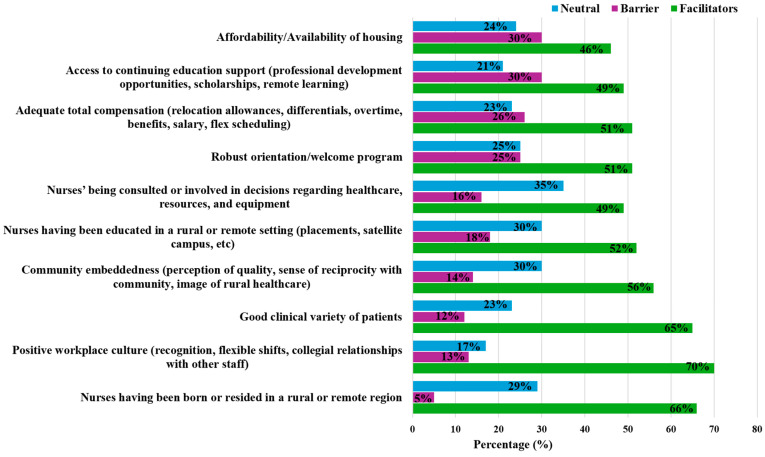
Perceived Facilitators of Nursing Graduates Entering Practice in Rural and Remote Areas of Manitoba.

**Figure 3 nursrep-15-00410-f003:**
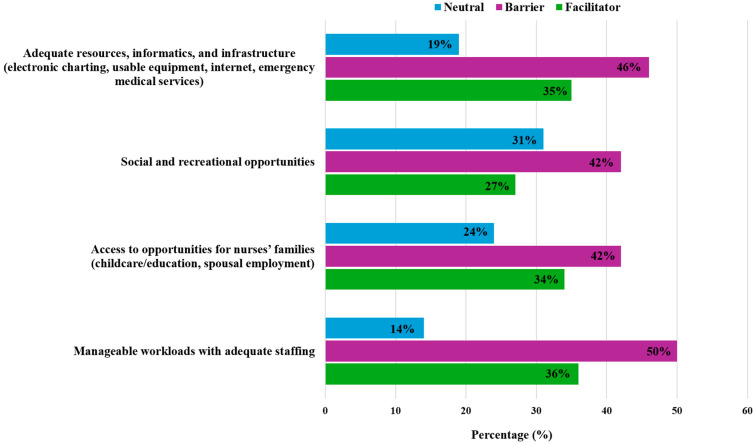
Perceived Barriers of Nursing Graduates Entering Practice in Rural and Remote Areas of Manitoba.

**Table 1 nursrep-15-00410-t001:** Demographic characteristics of study participants.

Demographic Characteristics	Quantitative Survey (*N* = 77)	Qualitative Interviews (*N* = 16)
	*n* (%)	*n* (%)
**Current Residence** Urban Rural Remote	4 (5.2%) 53 (68.8%) 20 (26.0%)	2 (12.5%) 12 (75.0%) 2 (12.5%)
**Marital Status** Single Married/Partnered Divorced/Widowed	32 (41.6%) 44 (57.1%) 1 (1.3%)	8 (50.0%) 8 (50.0%) -
**Years Living in Rural/Remote** 5 or less 6–10 11–15 More than 15 Other	28 (36.4%) 5 (6.4%) 6 (7.9%) 37 (48.0%) 1 (1.3%)	4 (25.0%) - 1 (6.3%) 10 (62.5%) -
**Employment Setting** Hospital Long Term Care Community Others	50 (64.9%) 4 (5.2%) 7 (9.1%) 16 (20.8%)	13 (81.2%) 1 (6.3%) 1 (6.3%) 1 (6.3%)
**Educational Level** LPN Diploma RN Degree Others	21 (27.3%) 56 (72.7%) -	5 (31.3%) 10 (62.5%) 1 (6.3%)
**Health Region** **Northern** Interlake Prairie Mountain Southern Winnipeg Others	18 (23.4%) 18 (23.4%) 13 (16.8%) 23 (29.9%) 2 (2.6%) 3 (3.9%)	4 (25.0%) 4 (25.0%) 4 (25.0%) 4 (25.0%) - -
**Length of License** Less than 12 months 13–24 months 25–36 months More than 36 months	23 (29.9%) 23 (29.9%) 22 (28.5%) 9 (11.7%)	3 (18.7%) 5 (31.3%) 5 (31.3%) 3 (18.7%)
**Educational Facility** Red River College Polytech Brandon University St. Boniface College University of Manitoba: Fort Gary University College of the North: University of Manitoba University College of the North: LPN Assiniboine Community College: LPN Other	14 (18.2%) 9 (11.7%) 3 (3.9%) 12 (15.6%) 10 (13.0%) 4 (5.2%) 12 (15.6%) 13 (16.8%)	3 (18.7%) 2 (12.5%) - 2 (12.5%) 1 (6.3%) - 3 (18.7%) 5 (31.3%)

**Table 2 nursrep-15-00410-t002:** Perceived Facilitators and Barriers for Rural and Remote Practice in Manitoba.

Rank	Variable	Observed Responses (*n* = 77)	*p*-Value	χ^2^ (*df* = 4)	α-FDR
	**Facilitators**				
1	**Q9**: Having a Positive Workplace Culture	54	0.001 *	42.73	0.003
2	**Q1**: Being Born & Raised in Rural/Remote areas	51	0.001 *	42.03	0.007
3	**Q12**: Having Variety of Clinical Patients	50	0.001 *	35.27	0.011
4	**Q14**: Community Embeddedness	43	0.001 *	22.45	0.014
5	**Q2**: Having Education in Remote/Rural areas	40	0.002 *	16.68	0.018
6	**Q8**: Being Involved in Health Decisions	39	0.002 *	16.53	0.021
7	**Q11**: Having Robust Welcome/Orientation	39	0.004 *	15.13	0.025
8	**Q4**: Having Adequate Compensation	38	0.006 *	14.56	0.029
9	**Q6**: Having Access to Educational Support	38	0.008 *	13.72	0.032
13	**Q3**: Having Affordable/Available Housing	32	0.261 NS	5.27	0.036
	**Barriers**				
10	**Q10**: Having Unmanageable Workloads	38	0.008 *	13.72	0.036
11	**Q5**: Having Limited Opportunities for Nurse Families	35	0.028 *	10.91	0.039
12	**Q7**: Having Limited Social & Recreational Activities	35	0.037 *	10.20	0.049
14	**Q13**: Having Inadequate Infrastructure	32	0.261 NS	5.27	0.05

* = significant (*p* < 0.05); NS = not significant (*p* ≥ 0.05).

**Table 3 nursrep-15-00410-t003:** Qualitative Themes and Sub-Themes Based on Participants’ Interview Responses.

Themes	Sub-Themes	Supporting Quotes
**Facilitators**		
Generalist Role	Patient Variety Being Versatile	“I think being rural, you really are like the jack of all trades.” RNPMRHA-1 “I like the diversity of patients, and I do like the acuity of my unit … it’s ever-changing. There’s always something new going on. I feel like I’m always learning every day.” RNNRHA-2 “I liked being able to come on to a shift and not knowing if we were going to have trauma or if we were going to have just minor treatment stuff all day.” RNNRHA-3
Autonomy	Independent Decision Making Increased Critical Thinking Skills Stabilizing Critical Patients	“The nurses will make a decision and then just update the doctor … 99% of the time they are good with it … send a fax, and great that works. Continue with a treatment plan. We have more autonomy.” LPNIERHA-1 “If someone comes in with an emergency, you as the nurse will have to either find the correct standing orders to go by or make decisions that need to be made even when the doctor is not there.” RNSHRHA-1
Rural Life	Rural Experiences Close Knit Community	“It’s fun living rurally. We’re all generally from the same area. One person’s doing something after work. We can all just go to their house to have like a bonfire or something.” RNSHRHA-4 “I am a rural person. I love being in the middle of nowhere. The great outdoors” RNPMRHA-3
Organizational Culture	Teamwork Tight Knit Hospital Atmosphere Supportive Management	“I feel like we are a tighter-knit community. You get to know the doctors a lot more. It’s a smaller team.” RNSHRHA-3 “I enjoy working with rural doctors, they’re a lot more friendly. Everyone is so friendly there. You felt very welcome. You were supported by your other colleagues.” RNPMHRHA-4 “Everywhere I’ve gone in the North has had such amazing managers that do what they can with what they have it really makes a difference.” RNNHRHA-4
**I. Barriers**		
Resources	Interdisciplinary Support Diagnostics, Labs, Imaging, Technology	“We have less resources, diagnostics and support in the North. We are always waiting for results from bigger centers.” RNNHRHA-2 “We don’t have healthcare aides or dietary.” RNSHRHA-2 “Like in rural sites, depending on how rural you go, I mean you could be the pharmacist, you could be the lab tech, you could be anything.” RNPMHRHA-3 “We do a lot of communicating via fax, which can be frustrating. Sometimes, it takes days to get a response from a doctor.” LPNIERHA-1
Staffing	Staffing Challenges Use of Agency Nursing	“Having staffing … working double time … being mandated it’s a major challenge of working in a rural site.” RNPMRHA-4 “We are short-staffed we always use agency nurses.” RNSHRHA-3 “The entire nursing team in a remote area is an agency.” RNNHRHA-4 “It does help with staffing. But I would say that there are probably more negatives than positives to using the agency … non-regular staff don’t know what the resident likes, they don’t know how their routines work, they do treatments and meds only.” RNPMRHA-3
Geography	Transport Challenges Weather Distance	“One major barrier right now in rural is transferring people to the city … EMS transport right now is so backlogged.” RNSHRHA-1 “I mean, specifically during the winter, there’s lots of issues even with just staffing, getting nurses to the hospitals can sometimes be very challenging … trying to transport anybody out EMS isn’t going to run in the middle of a snowstorm to an appointment.” RNSHRHA-2 “We are always waiting for flights out, and patients are not always stable.” RNNRHA-3
Expanded Role	Independent Decisions Leadership Roles	“One of the biggest barriers would just be having physicians on-site … putting us into those situations where we sometimes have to do things without them around or make decisions about what they need.” RNSHRHA-2 “I think I was in charge three months out of school or three months past when I got my license, and yeah, it felt very soon. The thing about that is at that time we had barebones staff, we barely had an RN cover each shift and our charge nurses are only our RN’s. So, um, yeah, I felt uncomfortable with being put in that situation.” RNNRHA-2 “You do more in rural. You have a larger scope; you need to take charge of your own learning. It is a huge learning curve.” RNPMRHA-3

RNNRHA—Registered Nurse Northern Regional Health Authority; RNSHRHA—Registered Nurse Southern Regional Health Authority; RNPMRHA—Registered Nurse Prairie Mountain Regional Health Authority; LPNIERHA—Licensed Practical Nurses Interlake Eastern Regional Health Authority.

## Data Availability

All data generated or analyzed during this study are included within the manuscript (or as a Appendix A). However, the original data (sets) collected during the study are not publicly available due to ethical concerns.

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
