# Peer review of "Facilitators, Barriers, and Educational Preparedness of Early-Career Nursing Graduates Entering Practice in Rural and Remote Areas: A Mixed-Method Study"

_nursrep, 2025, doi:10.3390/nursrep15110410_

Round 1

Reviewer 1 Report

Comments and Suggestions for Authors

The manuscript addresses a relevant topic and employs an interesting methodological design (mixed-methods). However, a substantial revision is required. As this is a mixed-methods study, it presupposes that the authors present results that integrate both data strands. At present, there is no demonstration of quantitative–qualitative integration, such as a joint display (a table combining quantitative results with qualitative interpretation). 

Title

  • Remove the locality.
  • Indicate “Early-Career Nursing Graduates”.
  • Exclude “A Sequential Explanatory”.

Keywords

  • Ensure that keywords follow controlled vocabulary (MeSH/Emtree) for better indexing.
  • Suggested terms: Rural Health Services, Nurse Retention.

Introduction

  • In the following passages, the authors provide different information on the same subject, which creates confusion for the reader:

Approximately 30% of the Canadian population lives in R&R settings, where only 10.8% of nurses are employed in these settings [2]. The number of nurses working in R&R areas in 2020 is estimated at 41,071 nurses, constituting only 10% of nurses in Canada [3]. Over the last decade, the proportion of nurses working in R&R areas has significantly declined from 11.1% in 2013 to 9.6% in 2022 [4]. In 2018, the rural nursing workforce in Manitoba was reported at 21.9%, illustrating a fast and steady decline in nurses working in these settings [5].

  • I suggest that the introduction should focus only on the contextualization of the topic: facilitators and barriers to entry into nursing practice in rural and remote (R&R) Canadian areas. In other words, present what is already known about the subject, both Canadian evidence and findings from other countries.
  • Information about Manitoba and its health system should be placed in the Methods, under study setting description.
  • The introduction should invite the reader to engage with the paper independently of where the study was conducted. This will enhance visibility for an international readership.
  • Although the problem was studied in Manitoba, the discussion should extrapolate to similar contexts.
  • The following description should also be moved to the Methods:

“While remote was defined as communities in Manitoba that do not have permanent road access, are more than a four-hour drive from a major rural hospital or have rail or fly-in access only. Northern is considered remote because of the distance to travel to an urban center.”

  • This statement contradicts the Statistics Canada definition and must be clarified:

“In the present study, rural was defined as communities with a core population of less than 10,000 people, where less than 50% of the population commutes to larger urban centers for work [15].”
vs.
“Statistics Canada defines rural areas as having a population density of less than 400 persons per square kilometre and fewer than 1000 people [1].”

  • Include a justification aligned with Canadian provincial/national workforce policies.

Methods

  • After the Study Design subsection, add Study Setting to describe the study context (population size, number of provinces, main city, number of large and medium hospitals). Including a map would enrich understanding.
  • Data Collection: consider providing the 14 NCAQ statements as supplementary material.
  • Quantitative phase:
    • Sampling and response rate: the number of invitations and response rate are not reported, making it impossible to assess selection bias. Provide this information.
    • Data analysis: did the authors use Fisher’s exact test when expected frequencies were low? Clarify.
  • Qualitative phase:
    • Could the interview guide be provided as supplementary material?
    • Was a pre-test conducted to validate the guide?
    • What was the average interview duration?
    • How was data saturation assessed? Please justify saturation or thematic adequacy.
    • Provide a detailed description of the qualitative data analysis process. Was software used? Who conducted the analysis?
  • Clarify whether the participants in the qualitative phase also participated in the quantitative phase.
  • Integration: although the authors state that the databases were “merged for analysis,” there is no concrete demonstration. At least one joint display is required.

Results

  • Acronyms in Table 1 should be explained in table notes.
  • Table 2 results should be presented more clearly—include columns for the outcomes being evaluated, showing results before the p-value.
  • Ensure uniform quality and formatting of Figures 1 and 2.
  • Insert a joint display (a table combining quantitative results with qualitative interpretation).

Discussion

  • Paragraphs are excessively long; division is necessary for readability.
  • The discussion must more explicitly highlight what this study adds beyond existing knowledge.
  • The Recommendations section should be rewritten in continuous text rather than as a list.
  • Limitations: currently only geographic limitation is mentioned. Please also address selection bias, statistical limitations (small n for tests with multiple categories), and potential response/social desirability bias. Discuss how these may affect interpretation.

Conclusion

  • Synthesize in 2–3 concise sentences the main findings and immediate implications.

References

  • Update references to include more recent studies.

Author Response

REVIEWER-1: 

Title

Comment: Remove the locality.

Authors’ response: Authors have revised the title as per the reviewers’ suggestion.

Comment: Indicate “Early-Career Nursing Graduates”.

Authors’ response: Authors have revised the title as per the reviewers’ suggestion.

Comment: Exclude “A Sequential Explanatory”.

Authors’ response: Authors have revised the title as per the reviewers’ suggestion.

Keywords

Comment: Ensure that keywords follow controlled vocabulary (MeSH/Emtree) for better indexing.

Authors’ response: Authors have revised the keywords as per the reviewers’ suggestion.

Comment: Suggested terms: Rural Health ServicesNurse Retention.

Authors’ response: Authors have revised the keywords as per the reviewers’ suggestion.

Introduction

Comment: In the following passages, the authors provide different information on the same subject, which creates confusion for the reader:

“Approximately 30% of the Canadian population lives in R&R settings, where only 10.8% of nurses are employed in these settings [2]. The number of nurses working in R&R areas in 2020 is estimated at 41,071 nurses, constituting only 10% of nurses in Canada [3]. Over the last decade, the proportion of nurses working in R&R areas has significantly declined from 11.1% in 2013 to 9.6% in 2022 [4]. In 2018, the rural nursing workforce in Manitoba was reported at 21.9%, illustrating a fast and steady decline in nurses working in these settings [5].”

Authors’ response: Authors have revised the statement for better readability and clarity as per the reviewers’ suggestion.

Comment: I suggest that the introduction should focus only on the contextualization of the topic: facilitators and barriers to entry into nursing practice in rural and remote (R&R) Canadian areas. In other words, present what is already known about the subject, both Canadian evidence and findings from other countries.

Authors’ response: Authors tried their best to address this given the limited time and word count concerns.

Comment: Information about Manitoba and its health system should be placed in the Methods, under study setting description.

Authors’ response: Authors have revised it as per the reviewers’ suggestion.

Comment: The introduction should invite the reader to engage with the paper independently of where the study was conducted. This will enhance visibility for an international readership.

Authors’ response: Agreed, and authors did what they could do in the given time.

Comment: Although the problem was studied in Manitoba, the discussion should extrapolate to similar contexts.

Authors’ response: Authors tried their best to address this given the limited time and word count concerns.

Comment: The following description should also be moved to the Methods:

“While remote was defined as communities in Manitoba that do not have permanent road access, are more than a four-hour drive from a major rural hospital or have rail or fly-in access only. Northern is considered remote because of the distance to travel to an urban center.”

Authors’ response: Authors have revised it as per the reviewers’ suggestion.

Comment: This statement contradicts the Statistics Canada definition and must be clarified:

“In the present study, rural was defined as communities with a core population of less than 10,000 people, where less than 50% of the population commutes to larger urban centers for work [15].” vs. “Statistics Canada defines rural areas as having a population density of less than 400 persons per square kilometre and fewer than 1000 people [1].”

Authors’ response: These are two different definitions. One from the introduction section, where authors attempted to explain how Statistics Canada defined the rural area; while the other definition is from the methods section, actually used by the authors in defining the rural areas in the current study keeping the Manitoba context in mind (while the Statistics Canada definition in general keeping the national context in mind, but not specific to Manitoba).

Methods

Comment: After the Study Design subsection, add Study Setting to describe the study context

Authors’ response: As per the reviewers’ suggestion, Study setting sub-section has been added.

Data Collection:

Comment: consider providing the 14 NCAQ statements as supplementary material.

Authors’ response: It has been provided as per the reviewers’ suggestion.

Quantitative phase:

Comment: Sampling and response rate: the number of invitations and response rate are not reported, making it impossible to assess selection bias. Provide this information.

Authors’ response: Authors have added the requested information as per the reviewers’ suggestion and can be reflected in the revised version of the manuscript.

Comment: Data analysis: did the authors use Fisher’s exact test when expected frequencies were low? Clarify.

Authors’ response: No, we did not conduct Fisher’s exact test, as authors settled on a sample size of 77 for practical reasons. Usually, increasing in ‘n’ will increase reliability. Since the reliability was already over the desired 0.88, the practical reasons won. So, a realistic sample size of 77 participants was recruited into the study.

Qualitative phase:

Comment: Could the interview guide be provided as supplementary material?

Authors’ response: It has been provided as per the reviewers’ suggestion.

Comment: Was a pre-test conducted to validate the guide?

Authors’ response: No, the modified survey was not piloted since the reliability of the M-NCAQ questionnaire was assessed by performing Cronbach’s alpha with 71 participants, resulting in an excellent internal consistency value of 0.88.

Comment: What was the average interview duration?

Authors’ response: On average, each interview was about an hour to 2 hours approx.

Comment: How was data saturation assessed? Please justify saturation or thematic adequacy.

Authors’ response: Stopped data collection when the interviewer felt new data/information no longer provides new insights or themes. Further, data collection was stopped when themes started to repeat consistently across multiple data points. This has been added to the revised version of the manuscript.

Comment: Provide a detailed description of the qualitative data analysis process. Was software used? Who conducted the analysis?

Authors’ response: A qualitative analytical approach, directed content analysis was utilized in this study. Inductive analytical techniques were incorporated including repeat immersion in the data during the coding, classifying and creating linkages in the data. The transcripts were then taped with verbatim via an online platform and all the transcripts were reviewed, coded and analyzed by both the primary RA and PI. It was detailed in 2.3 Participant Recruitment sub-section under methods section.

Comment: Clarify whether the participants in the qualitative phase also participated in the quantitative phase.

Authors’ response: Yes, the participants in the qualitative phase also participated in the quantitative phase. It was also stated in 2.3 Participant Recruitment sub-section under methods section.

Comment: Integration: although the authors state that the databases were “merged for analysis,” there is no concrete demonstration. At least one joint display is required.

Authors’ response: This is a sequential mixed-method design but not convergent to integrate and compare both QUANT results with QUAL results. Usually, integration of the quantitative results with the qualitative results take place with convergent design. [Convergent mixed methods involve collecting quantitative and qualitative data simultaneously and then comparing the findings to provide a comprehensive understanding, while sequential mixed methods involve collecting data in separate phases, with the results from the first phase informing or building upon the second]

Results

Comment: Acronyms in Table 1 should be explained in table notes.

Authors’ response: Revised as per reviewers’ suggestions.

Comment: Table 2 results should be presented more clearly—include columns for the outcomes being evaluated, showing results before the p-value.

Authors’ response: The facilitator and barrier questions presented in table 2 are revised for better understanding and clarity. Authors are not sure what “include columns for the outcomes being evaluated, showing results before the p-value” means.

Comment: Ensure uniform quality and formatting of Figures 1 and 2.

Authors’ response: Revised as per reviewers’ suggestions.

Comment: Insert a joint display (a table combining quantitative results with qualitative interpretation).

Authors’ response: As stated above, this is a sequential mixed-method design but not convergent to integrate and compare both QUANT results with QUAL results. Usually, integration of the quantitative results with the qualitative results take place with convergent design.

Discussion

Comment: Paragraphs are excessively long; division is necessary for readability.

Authors’ response: Revised as per reviewers’ suggestions.

Comment: The discussion must more explicitly highlight what this study adds beyond existing knowledge.

Authors’ response: Authors attempted their best to address this comment keeping the word count and time provided in mind.

Comment: The Recommendations section should be rewritten in continuous text rather than as a list.

Authors’ response: With due respect, authors believe that reporting recommendation in the point/list form would make it easier for the readers rather than putting them in a paragraph format.

Comment: Limitations: currently only geographic limitation is mentioned. Please also address selection bias, statistical limitations (small n for tests with multiple categories), and potential response/social desirability bias. Discuss how these may affect interpretation.

Authors’ response: Authors believe there is not much chance for the selection bias to occur with the type of participants included in the study. Also, we firmly believe a sample of size of 77 is decent enough to generalize the conclusions to all the early career nursing graduates at least within the Canadian context. However, there might be a possibility for response/social desirability bias, which has been added as a potential limitation in the revised version of the manuscript.

Conclusion

Comment: Synthesize in 2–3 concise sentences the main findings and immediate implications.

Authors’ response: Revised as per reviewers’ suggestions.

References

Comment: Update references to include more recent studies.

Authors’ response: Authors tried their best to update the references wherever possible and needed.

Reviewer 2 Report

Comments and Suggestions for Authors

Hello Team

An excellent paper thank you for inviting me to review your study. I have provided comments throughout the paper some are my own beliefs please disregard if they don't align. For example I believe the word 'also' is not required or that undergraduate nurse training is general not specialized. However there are a number of comments that could strengthen the evidence provided. In Australia the use of Telehealth to support to nurses is valuable this decreases the risk of working outside the scope of practice. It was interesting to note that you paper mentioned mentorship between the doctors and nurses but did not highlight preceptorship or mentorship between nurses. 

Author Response

REVIEWER-2:

Comment: How many RN’s and LPN’s?

Authors’ response: Authors addressed the comment, and it can be reflected in the revised version of the manuscript.

Comment: What constitutes a rural setting in Canada?

Authors’ response: Authors clearly stated what constitutes a rural setting in Canada in the second statement of the introduction section.

Comment: What does this percentage actual mean? Is it 21.9% of the overall population of Canada OR rural and remote areas OR of the expected workforce for rural and remote?

Authors’ response: Authors revised the statement for better readability and understanding as per the reviewers’ comment.

Comment: Higher rates of what?

Authors’ response: Authors revised the statement for better readability and understanding as per the reviewers’ comment.

Comment: This is vague what is the percentage?

Authors’ response: Authors revised the statement accordingly.

Comment: What is an RPN?

Authors’ response: Authors addressed the comment, and it can be reflected in the revised version of the manuscript.

Comment: This survey was not provided as a supplementary document.

Authors’ response: Authors addressed the concern, and it can be reflected in the supplementary files of revised manuscript.

Comment: Was this modified survey piloted before it was distributed?

Authors’ response: No, the modified survey was not piloted since the reliability of the M-NCAQ questionnaire was assessed by performing Cronbach’s alpha with 71 participants, resulting in an excellent internal consistency value of 0.88.

Comment: Have you predicted an outcome before the study commenced?

Authors’ response: No, the study’s outcome was not predicted before the study commenced. Authors have revised the statement for better readability and clarity as per the reviewers’ suggestion.

Comment: How was this analysed - did you conduct quantitative content analysis?

Authors’ response: Yes, content analysis was performed. Authors have revised the statement for better readability and clarity as per the reviewers’ suggestion.

Comment: What does this mean?

Authors’ response: Authors have revised the statement for better readability and clarity as per the reviewers’ suggestion.

Comment: Meaning - a mix of the above?

Authors’ response: Authors have revised the statement for better readability and clarity as per the reviewers’ suggestion.

Comment: I though you were not including this group in the study? 

Authors’ response: Yes, we did not include any RPNs in the quantitative survey. However, one participated in the qualitative interview. Authors have revised the statement for better readability and clarity as per the reviewers’ suggestion.

Comment: No idea what this stands for...

Authors’ response: Authors have revised the statement for better readability and clarity as per the reviewers’ suggestion.

Comment: What is UM?

Authors’ response: Authors have revised the statement for better readability and clarity as per the reviewers’ suggestion.

Comment: This seems repetitive.

Authors’ response: Authors have revised the statement for better readability and clarity as per the reviewers’ suggestion.

Comment: What is the p value?

Authors’ response: Authors have revised the statement for better readability and clarity as per the reviewers’ suggestion.

Comment: What about theatre nursing or do you consider this as part of surgical?

Authors’ response: Here in Manitoba, Canada we do not call it as theatre nursing. If you are asking about OR (operating room) nursing, we did not have any participants from this category included in the study.

Comment: This interrupted the flow - consider rewording - In Manitoba the sample included most of the…

Authors’ response: Authors have revised the statement for better readability and clarity as per the reviewers’ suggestion.

Comment: Thought bubble when reading this …..Undergraduate programs to offer a rural placement or recruitment to look for students who have completed a rural placement in their undergraduate programs…

Authors’ response: Good point. Authors have revised the statement for better readability and clarity as per the reviewers’ suggestion.

Comment: Is Telehealth available?

Authors’ response: Yes, Telehealth is available in Canada but primarily used for medical appointments with specialists.

Comment: How many agency staff did you interview this was not made clear in your demographics?

Authors’ response: A total of 5 Agency nurses were interviewed qualitatively; where 2 lived in urban and 3 lived in rural. Authors have revised the statement for better readability and clarity as per the reviewers’ suggestion.

Comment: Isn’t this taught in the undergraduate degree?

Authors’ response: No, here in Manitoba, Canada ECG was not taught in the undergraduate degree.

Comment: How big was the hospital would this be equivalent to be in charge of a ward in a metropolitan setting?

Authors’ response: It is not equivalent to a ward in the metropolitan setting; it is somewhere close to <100 beds hospital.

Comment: Is there Telehealth to support the LPN - would they complete an incident report for unsafe practice?

Authors’ response: No, TH is primarily used for medical appointments with specialists

Comment: Interesting I wonder if this is due to undergraduate training? As this was a large area of research a number of years ago.

Authors’ response: Authors have revised the statement for better readability and clarity as per the reviewers’ suggestion.

Comment: Do you mean decline?

Authors’ response: Authors have revised the statement for better readability and clarity as per the reviewers’ suggestion.

Comment: This would jeopardize their registration or licence.

Authors’ response: Authors have revised the statement for better readability and clarity as per the reviewers’ suggestion.

Comment: I wonder if it would be better to just write ‘took’ instead? Keep in mind the audience you want to engage.

Authors’ response: Authors have revised the statement for better readability and clarity as per the reviewers’ suggestion.

Comment: So is it outside of their scope or to the full extent?

Authors’ response: Authors have revised the statement for better readability and clarity as per the reviewers’ suggestion.

Comment: Consider which surname comes first and check your reference list - when she divorced I believe she dropped the Boychuk name.

Authors’ response: Authors have revised the statement for better readability and clarity as per the reviewers’ suggestion.

Comment: And remove what content?

Authors’ response: Authors have revised the statement for better readability and clarity as per the reviewers’ suggestion.

Comment: I am not sure a tour would be enough however placements are great.

Authors’ response: Authors have revised the statement for better readability and clarity as per the reviewers’ suggestion.

Comment: Would this be cost effective?

Authors’ response: We at the RRC already have a rural program and have 2 satellite sites and infrastructure and would like to expand more in the coming years.

Comment: What do you mean? 

Authors’ response: Authors have revised the statement for better readability and clarity as per the reviewers’ suggestion.

Reviewer 3 Report

Comments and Suggestions for Authors

Dear Authors,

First of all, I would like to congratulate you on presenting a manuscript that addresses a highly relevant issue in the field of nursing: the educational preparedness and the barriers/facilitators experienced by nursing graduates entering practice in rural and remote areas of Manitoba. The study provides timely and necessary evidence to guide the training of future professionals and to inform retention strategies in contexts characterised by workforce shortages.

Below, I share comments and suggestions for improvement, organised by section:

Introduction

  • It would be helpful to emphasise more clearly the specific knowledge gap in Manitoba that motivates the study.

Methodology

This section is sound, yet it requires further detail to strengthen transparency and reproducibility:

  • Quantitative instrument: the modified NCAQ questionnaire is mentioned, but the process of item selection and reduction from 50 to 14 is not explained in depth. It would be valuable to specify the modification criteria and provide examples of items removed or adapted.
  • Validity: although a satisfactory Cronbach’s alpha is reported, there is no discussion of whether content or construct validation was undertaken after the questionnaire was modified. Including this information, or acknowledging it as a limitation, would be advisable.
  • Sample: the description of participants is clear, but a stronger justification of the quantitative sample size is needed.
  • Qualitative phase: further detail on the analytical process would be beneficial (e.g. software used, how discrepancies were resolved).
  • Ethical considerations: these are well described, but the explanation of how confidentiality was ensured in small communities—where indirect identification is more likely—could be reinforced.

Results

  • The quantitative results are clearly presented but would benefit from further graphical synthesis (e.g. more visual tables or figures).
  • In the qualitative results, the selected quotations are rich and representative; however, balancing the number of positive and negative examples would help to avoid interpretative bias.

Discussion

  • The discussion connects the findings appropriately with existing literature, although in some parts results already described are repeated. Condensing these would allow greater space for critical interpretation.
  • It would be advisable to expand on the applicability of the findings to other rural contexts beyond Manitoba and to reflect more on the limitations of generalisability.
  • Although the importance of rural placements in training is mentioned, it would be valuable to discuss which specific educational strategies could be implemented in collaboration with universities and healthcare services.

Author Response

REVIEWER-3:

Introduction

Comment: It would be helpful to emphasize more clearly the specific knowledge gap in Manitoba that motivates the study.

Authors’ response: As per the reviewers’ suggestion, a couple of points that highlight the knowledge gap and importance of the current study have been added to the revised version of the manuscript. For instance, “soaring vacancy rates of 6.4% (Canada’s national average), whereas it is close to 10% in Manitoba (far greater than the national average)”

Methodology

This section is sound, yet it requires further detail to strengthen transparency and reproducibility:

Comment: Quantitative instrument: the modified NCAQ questionnaire is mentioned, but the process of item selection and reduction from 50 to 14 is not explained in depth. It would be valuable to specify the modification criteria and provide examples of items removed or adapted.

Authors’ response: Authors have added the suggested information and can be reflected in the revised version of the manuscript.

Comment: Validity: although satisfactory Cronbach’s alpha is reported, there is no discussion of whether content or construct validation was undertaken after the questionnaire was modified. Including this information, or acknowledging it as a limitation, would be advisable.

Authors’ response: Yes, quantitative content analysis was performed. Authors have revised the statement for better readability and clarity as per the reviewers’ suggestion. Also, a qualitative analytical approach, directed content analysis was utilized in this study. Inductive analytical techniques were incorporated including repeat immersion in the data during the coding, classifying and creating linkages in the data. The transcripts were then taped with verbatim via an online platform and all the transcripts were reviewed, coded and analyzed by both the primary RA and PI. It was detailed in 2.3 Participant Recruitment sub-section under methods section.

Comment: Sample: the description of participants is clear, but a stronger justification of the quantitative sample size is needed.

Authors’ response: Authors settled on a sample size of 77 for practical reasons. Usually, increasing in ‘n’ will increase reliability. Since the reliability was already over the desired 0.88, the practical reasons won. Also, typically, 8-10 participants per test item are suggested. Therefore, a realistic sample size of 77 participants was recruited into the study.

Comment: Qualitative phase: further detail on the analytical process would be beneficial (e.g. software used, how discrepancies were resolved).

Authors’ response: More details have been added to 2.3 Participant Recruitment sub-section under methods section.

Comment: Ethical considerations: these are well described, but the explanation of how confidentiality was ensured in small communities—where indirect identification is more likely—could be reinforced.

Authors’ response: All the data collected from the participants were anonymized and de-identified. Authors have now added a statement around this in the strengths and limitations section of the revised manuscript.

Results

Comment: The quantitative results are clearly presented but would benefit from further graphical synthesis (e.g. more visual tables or figures).

Authors’ response: Quantitative results have been represented both in the form of tables and figures.

Comment: In the qualitative results, the selected quotations are rich and representative; however, balancing the number of positive and negative examples would help to avoid interpretative bias.

Authors’ response: We respect the reviewers’ comment; however, since the methodology we employed is the sequential explanatory design, it is not possible to separate and maintain a balance between positive and negative qualitative quotes. It must be in a sequence.

Discussion

Comment: The discussion connects the findings appropriately with existing literature, although in some parts results already described are repeated. Condensing these would allow greater space for critical interpretation.

Authors’ response: Authors tried their best to address this comment. Now in the revised version of the manuscript, authors revised the discussion section for better readability and clarity.

Comment: It would be advisable to expand on the applicability of the findings to other rural contexts beyond Manitoba and to reflect more on the limitations of generalizability.

Authors’ response: Authors have now added a statement in the limitations section regarding the generalizability of the study finding to other rural communities outside of Manitoba, Canada.

Comment: Although the importance of rural placements in training is mentioned, it would be valuable to discuss which specific educational strategies could be implemented in collaboration with universities and healthcare services.

Authors’ response: Authors attempted to address this suggestion partly in the recommendations section, which were drawn directly from the study participants during the qualitative interview process. Besides, this suggestion has now been added as one of the future directions of the study.

Round 2

Reviewer 1 Report

Comments and Suggestions for Authors

Dear Authors,

The contextualization of the topic – Facilitators, Barriers, and Educational Preparedness of Early-Career Nursing Graduates Entering Practice in Rural and Remote Areas – remains limited to the Canadian context (most of the references used are from Canada).
It is expected that the authors present the state of the art on the subject, that is, extend the discussion beyond the Canadian setting to include international evidence and perspectives.

The description of the methodology has improved; however, some points still need clarification.
When asked whether “Was a pre-test conducted to validate the guide?”, the authors must include this information in the text, specifically regarding the pre-test of the interview guide used in the qualitative study. Please refer to the COREQ checklist from the EQUATOR Network for guidance.

Regarding the duration of the interviews, the manuscript states that they lasted between one and two hours. Could the authors confirm whether this information is accurate?

Concerning the use of Fisher’s exact test, the response provided does not address my original question. Fisher’s exact test is used to determine whether there is a significant association between two categorical variables and is typically applied as an alternative to the chi-square test of independence when one or more cell counts in a 2×2 table are less than five.

In the quantitative study, the sample consists of 77 participants. I believe that applying inferential statistical tests with such a small sample size compromises the validity of the results, leading to weak or unreliable conclusions. Therefore, it would be more appropriate to treat this as a descriptive study, properly presenting, in the columns (Table 2), the outcome variables and the number of observations for each outcome.

Regarding the study limitations, the authors state that “we firmly believe a sample size of 77 is decent enough to generalize the conclusions to all the early career nursing graduates at least within the Canadian context.” I disagree with this assertion and strongly recommend that the study present only descriptive data, without performing inferential analysis.

Author Response

Authors’ Response to Reviewers’ Comments (Round-2)

Comment: The contextualization of the topic – Facilitators, Barriers, and Educational Preparedness of Early-Career Nursing Graduates Entering Practice in Rural and Remote Areas – remains limited to the Canadian context (most of the references used are from Canada). It is expected that the authors present the state of the art on the subject, that is, extend the discussion beyond the Canadian setting to include international evidence and perspectives.

Authors’ response: The authors have revised the manuscript as per the reviewers’ suggestions. A couple of paragraphs (six recent studies) at the end of the discussion section have been added, discussing the most recent literature outside the Canadian context.

Comment: The description of the methodology has improved; however, some points still need clarification. When asked whether “Was a pre-test conducted to validate the guide?”, the authors must include this information in the text, specifically regarding the pre-test of the interview guide used in the qualitative study. Please refer to the COREQ checklist from the EQUATOR Network for guidance.

Authors’ response: The authors have revised the manuscript as per the reviewers’ suggestions. The suggested information has been mentioned in the methods section under 2.2 Data Collection of methods section.

Comment: Regarding the duration of the interviews, the manuscript states that they lasted between one and two hours. Could the authors confirm whether this information is accurate?

Authors’ response: Of course, the information that has been disclosed is accurate and correct.

Comment: Concerning the use of Fisher’s exact test, the response provided does not address my original question. Fisher’s exact test is used to determine whether there is a significant association between two categorical variables and is typically applied as an alternative to the chi-square test of independence when one or more cell counts in a 2×2 table are less than five.

Authors’ response: No, unfortunately, we did not use Fisher’s exact test.

Comment: In the quantitative study, the sample consists of 77 participants. I believe that applying inferential statistical tests with such a small sample size compromises the validity of the results, leading to weak or unreliable conclusions. Therefore, it would be more appropriate to treat this as a descriptive study, properly presenting, in the columns (Table 2), the outcome variables and the number of observations for each outcome.

Authors’ response: The Authors have added ‘observed responses’ as a separate column in Table 2 as per the reviewer's suggestion. Besides, rather than entirely removing the tests along with their results from Table 2, as they were already performed and reported, the authors highlighted this as a limitation and cautioned readers to read the findings carefully, keeping the small sample size in mind.

Comment: Regarding the study limitations, the authors state that “we firmly believe a sample size of 77 is decent enough to generalize the conclusions to all the early career nursing graduates at least within the Canadian context.” I disagree with this assertion and strongly recommend that the study present only descriptive data, without performing inferential analysis.

Authors’ response: The Authors have removed the above statement and highlighted having a sample size of 77 as a limitation in performing advanced inferential statistics.

Reviewer 3 Report

Comments and Suggestions for Authors

Accept in present form

Author Response

.